# Variation of soil hydraulic properties with alpine grassland degradation in the Eastern Tibetan Plateau

Tao Pan[1,3], Shuai Hou[1,2], Shaohong Wu[1], Yujie Liu[1], Yanhua Liu[1], Xintong Zou[1,2], Anna Herzberger[3], Jianguo Liu[3]

[1]Key Laboratory of Land Surface Pattern and Simulation, Institute of Geographic Sciences and Natural Resources Research, Chinese Academy of Sciences, Beijing 100101, China
[2]University of the Chinese Academy of Sciences, Beijing 100049, China
[3]Center for Systems Integration and Sustainability, Department of Fisheries and Wildlife, Michigan State University, East Lansing, MI 48824, USA

*Correspondence to*: Yujie Liu (liuyujie@igsnrr.ac.cn)

**Abstract.** Ecosystems in alpine mountainous regions are vulnerable and easily disturbed by global environmental change. Alpine swamp meadow, a unique grassland type in the eastern Tibetan Plateau that provides important ecosystem services to the upstream and downstream regions of international rivers of Asia and other parts of the world, is undergoing severe degradation, which can dramatically alter soil hydraulic properties and water cycling processes. However, the effects of alpine swamp meadow degradation on soil hydraulic properties and the corresponding influencing mechanisms are still poorly understood. In this study, soil moisture content (SMC), field capacity (FC) and saturated hydraulic conductivity (Ks) together with several basic soil properties under lightly degraded (LD), moderately degraded (MD) and severely degraded (SD) alpine swamp meadow were investigated; the variations in SMC, FC and Ks with alpine swamp meadow degradation and their dominant influencing factors were analysed. The results showed that SMC and FC decreased consistently from LD to SD, while Ks decreased from LD to MD and then increased from MD to SD, following the order of LD > SD > MD. Significant differences in soil hydraulic properties between degradation degrees were found in the upper soil layers (0–20 cm), indicating that the influences of degradation were most pronounced in the topsoils. FC was positively correlated with capillary porosity, water-stable aggregates, soil organic carbon and silt and clay content; Ks was positively correlated with non-capillary porosity (NCP). Relative to other soil properties, soil porosity is the dominant factor influencing FC and Ks. Capillary porosity explained 91.1% of total variance in FC, and NCP explained 97.3% of total variance in Ks. The combined effect of disappearing root activities and increasing sand content was responsible for the inconsistent patterns of NCP and Ks. Our findings suggest that alpine swamp meadow degradation would inevitably lead to reduced water holding capacity and rainfall infiltration. This study provides a more comprehensive understanding of the soil hydrological effects of vegetation degradation. Further hydrological modelling studies in the Tibetan Plateau and similar regions are recommended to understand the effects of degraded alpine swamp meadow on soil hydraulic properties.

**Key words: soil hydraulic properties; field water capacity; saturated hydraulic conductivity; alpine swamp meadow degradation; influencing factors; Tibetan Plateau**

## 1 Introduction

Soil moisture plays a critical role in land surface processes and hydrological cycles. It not only directly participates in soil hydrological processes but also influences vegetation growth and even modifies weather processes and local climate (Legates et al., 2011; Shein, 2010; Vereecken et al., 2015). Field capacity (FC) and saturated hydraulic conductivity (Ks) are two key soil hydraulic properties that jointly affect soil water storage, transmission and distribution (Cassel et al., 1986; Marshall et al., 2014). Knowledge of how FC and Ks vary and of their influencing factors is essential for better understanding of soil hydrological processes. FC and Ks are also key parameters in most hydrological, climate and land surface models (Boluwade et al., 2013; Reszler et al., 2016; Tatsumi et al., 2015). Therefore, understanding the effects of vegetation changes on FC and Ks is necessary for model parameterization and reducing the uncertainty of simulations (Sun et al., 2016).

Soil hydraulic properties are highly heterogeneous both spatially and temporally and could respond swiftly to external changes and disturbances (Ma et al., 2016; Strudley et al., 2008). FC and Ks are mainly influenced by vegetation, soil (Pachepsky et al., 2015), topography (Leij et al., 2004), climate (Jarvis et al., 2013) and human activities (Mubarak et al., 2009; Palese et al., 2014). In recent years, vegetation degradation has been widespread because of natural environmental changes and anthropogenic influences. Efforts have been devoted to revealing the effects of vegetation degradation on soil hydraulic properties across scales and ecosystem types. For forest, Lal (1996), Niemeyer et al. (2014) and Zimmermann et al. (2008) analysed the variations in Ks along gradients of disturbance and confirmed the increasing trend of Ks with forestation processes. Krummelbein et al. (2009) investigated the effects of grazing intensity on soil hydraulic properties and revealed the variations in soil porosity and soil retention characteristics in Inner Mongolian grasslands. Recently, Fu et al. (2015) and Price et al. (2010) explored the variations in Ks with land use changes and demonstrated the decrease in Ks and the subsequent increase in overland flow resulting from deforestation or cultivation. Despite these advances, existing studies mostly focused on low altitude areas. There are still many other regions where the effects of vegetation degradation on soil hydraulic properties are inadequately studied (Vereecken, et al., 2015). This is highlighted in remote areas such as alpine mountainous regions; these are cold and adverse environments, where fieldwork is time-consuming and extremely labour intensive (Bernhardt, et al., 2014; Wang, et al., 2012).

Alpine mountainous regions are widely distributed around the world, such as the Rocky Mountain Range in North America, the Andean mountain range in South America, the Alps in Europe, Mt. Kilimanjaro in Africa, the Himalayans in Tibet, and Mt. Fuji in Japan. These regions are often the headwaters of great rivers and supply a large amount of water to the lower

reaches (Bernhardt et al., 2014; Kormann et al., 2015). Alpine grassland is one of the main ecosystem types that plays an important role in water supply service of these regions. Because of the extreme environmental conditions of alpine mountainous regions, the alpine ecosystems are very vulnerable to environmental change. Because of global climate warming and human disturbances, alpine grasslands are experiencing intense changes generally leading to degradation,

which can have great influence on soil hydrological processes. Therefore, understanding the responses of soil hydraulic properties to alpine grassland variations is of great importance (Laghari et al., 2012). For example, in central Nepal, south of the Himalayas, Ghimire et al. (2014) and Prasad et al. (2013) investigated variations in Ks with land use changes and found that reforestation of degraded pasture could substantially increase Ks in surface and subsurface layers. Leitinger et al. (2010) studied FC and soil infiltration rate of alpine pasture in the Alps and found that grazing significantly decreased FC and soil

infiltration rate in the 0–20 cm layer, and thus altered surface runoff generation. In the Tibetan Plateau, Wang et al. (2012) confirmed that changes in alpine grassland vegetation cover will greatly alter infiltration processes and hill slope runoff coefficients. Although these studies contributed to a better understanding of the effect of alpine ecosystem variations on soil hydraulic properties, they primarily addressed the change patterns of hydraulic properties. Moreover, it has been confirmed that soil physical and chemical properties, such as soil porosity, texture and organic matter content, are closely connected

with soil hydraulic properties (Fu et al., 2015; Jarviset al., 2013; Strudley et al., 2008). However, studies about the quantitative relationships between soil hydraulic properties and physiochemical properties are still relatively limited.

Widely known as 'the third pole' and the highest place in the world, the Tibetan Plateau is the headwaters region of the Yangtze, Yellow and Mekong rivers, which are the world's third, fifth and seventh longest rivers, respectively. The hydrological cycling of this region has great influence on the energy and water processes of eastern Asia. Because of climate

change, overgrazing, human activities and rodents, alpine meadow on the Tibetan Plateau has been severely degraded (Wang et al., 2007). Although hydrological effects of alpine meadow degradation over the Tibetan Plateau have been extensively explored (Li et al., 2012; Wang et al., 2007; Wang et al., 2010; Zeng et al., 2013), large discrepancies still exist in the conclusions obtained. For example, Zeng et al. (2013) analysed the effect of alpine meadow degradation on soil hydraulic properties using tension infiltrometers in the centre of the Tibetan Plateau and found that Ks generally decreased with

degradation for both 0–10 and 40–50 cm layers; Wei et al. (2010) investigated the impact of alpine meadow degradation on soil infiltration by double-ring infiltrator in the eastern Tibetan Plateau and reported that Ks of the 0–20 cm layer decreased initially and then increased. Wang et al. (2007) measured Ks of alpine steppe and alpine meadow at the source of the Yangtze River and found that Ks of the surface layers increased significantly with degradation. Yi et al. (2012) reported that FC of the 0–20 cm layer was highest for non-degraded alpine meadow while Li et al. (2012) found that the highest value of

FC occurred with light degradation. These inconsistencies showed the high variability of soil hydraulic properties on the Tibetan Plateau and called for further investigations. To date, however, comparison of different change patterns of previous studies and the underlying mechanisms are rarely reported. As the main grassland type in the eastern Tibetan Plateau, alpine swamp meadow has unique terrestrial–aquatic soil and vegetation characteristics (Shang et al., 2013; Zedler et al., 2005).

However, little attention has been paid to the effects of alpine swamp meadow degradation on soil hydraulic properties and the influencing mechanisms, which might serve as a bottleneck for a thorough understanding of the hydrological effects of alpine grassland degradation on the Tibetan Plateau.

In this study, we measured Ks and FC along with several key basic soil properties based on a series of experimental sites that represent the degradation process of alpine swamp meadow in the eastern Tibetan Plateau. Several statistical methods such as redundancy analysis were used to quantitatively analyse the variation in soil hydraulic properties and its influencing factors. This study aimed to (1) investigate changes in FC and Ks associated with degradation, and (2) analyse the dominant factors and reveal the influencing mechanism of degradation on FC and Ks for alpine swamp meadow.

## 2 Material and methods

### 2.1 Site description

The experimental field (102°12'45" E, 33°46'28" N, 3435 m above sea level) is located in the Zoige Wetland in the east of the Tibetan Plateau (Fig. 1a). This region contains the largest area of alpine swamp in China and is the main recharge area of the Yellow River (Bai et al., 2013). In recent decades, however, a large proportion of wetland area has been converted from swamp to meadow, which in some cases has resulted in desertification (Hu et al., 2015). The mean daily air temperature is 1.2°C, ranging from −10.7°C in January to 11.7°C in July, and the average annual precipitation is 620 mm, 85% of which falls during the summer. The main vegetation is *Kobresia*-dominated alpine meadow (e.g., *Kobresia tibetica*, *Blysmus sinocompressus*, *Carex muliensis* and others) and the corresponding soil is silt loam, an alpine meadow soil type (Huo et al., 2013).

The experimental field is relatively flat with no perceivable slope and an elevation difference of 20 m between the highest and lowest points (Fig. 1b). Because of variation in grazing intensity, rodent activities and topographic conditions, patches of grassland from initial degradation to almost completely barren have emerged across the field, making it possible to choose sites in various degrees of degradation in small areas without large-scale soil spatial heterogeneity confounding the results.

Based on the survey of herbage growth and dominant species, a total of nine investigated sites representing various degrees of degradation were selected along the strip of the enclosed experimental field (Fig. 1b) using a strategy of space-for-time substitution (Zeng et al., 2013). To assess the degree of degradation of each site, several key vegetation characteristics including total vegetation coverage (VC), dominant species, number of species, aboveground biomass (MA), and belowground biomass (MB) were determined in mid–late July, 2014. Average field plant height was recorded at 10–15 cm.

For the classification of alpine degradation, various qualitative and semi qualitative indicators are present in the literature (Gao et al., 2010; Wang et al., 2007; Zeng et al., 2013). In this study, we chose VC, dominant species and number of species as indicators of degradation, and the nine sites were classified into three groups: lightly degraded (LD), moderately degraded (MD), and severely degraded (SD), corresponding to sites 1, 2 and 4, sites 5, 6 and 8 and sites 3, 7 and 9, respectively.

5 Characteristics of the three degradation degrees of alpine meadow are shown in Table 1, and MA and MB of each degree are shown in Fig. 2.

## 2.2 Soil sampling and measurements

Both disturbed and undisturbed soil samples were obtained from 0 to 80 cm depths at 10 cm intervals at three points randomly distributed in each investigated site mentioned above. Disturbed samples were collected using a soil auger, and

10 samples of the same layer were thoroughly mixed and then air-dried. After being sieved by 2 mm and 0.15 mm mesh, the composite samples were stored in plastic bags and transported to the laboratory for analysis. Soil organic carbon (SOC) was determined by dichromate oxidation with an external heat source (also cited as Walkley-Black wet combustion method; Nelson et al., 1996); 1–2 mm water-stable aggregates (WSA) were measured using a routine wet-sieve method with mechanical sieving procedure described by ISSAS (1978); soil particle composition (sand >0.05 mm, silt 0.002–0.05 mm,

15 and clay <0.002 mm) was analysed by wet sample dispersion and laser diffraction method using a laser-scattering particle analyser (Microtrac S3500, Microtrac Inc., USA; Cooper et al., 1984; Zhang, 2014).

Undisturbed samples were collected using cylinder cores (50.46 mm in diameter and 50 mm in height) to determine soil physical and hydraulic properties including bulk density (BD), capillary porosity (CP), non-capillary porosity (NCP), field water capacity (FC) and saturated hydraulic conductivity (Ks). In the laboratory, all these parameters were determined in proper sequence with the water suction method (Fu et al., 2015). First, the cylinder cores were dipped in 5 mm depth water to absorb water through capillary action for roughly 8 h before a constant weight was reached; the corresponding weights were recorded as m1. Second, the cores were soaked in 4.8 cm depth water for approximately 24 h until saturated, and the respective weights were recorded as m2. Third, soil samples were put on dry sand for 48 h and the resulting weights were recorded as m3. Subsequently, cylinder cores were linked to a Mariotte's bottle to measure Ks using the constant head method based on Darcy's Law (Klute et al., 1986). Finally, the cores were oven-dried at 105°C for approximately 24 h and the weights were recorded as m4. No perceivable swelling was detected for all the cores during the soaking process, and the parameters were calculated by the following formulas:

$$BD = \frac{m4}{V} \qquad (1)$$

$$CP = \frac{m1 - m4}{\rho \cdot V} \qquad (2)$$

$$NCP = \frac{m2 - m1}{\rho \cdot V} \qquad (3)$$

$$FC = \frac{m3 - m4}{BD \cdot V} \qquad (4)$$

$$Ks = \frac{10 \cdot Q \cdot L}{A \cdot \Delta H \cdot t} \qquad (5)$$

where V is the volume of the cylinder core ($100 cm^3$); $\rho$ is the water density ($1 \ g \ cm^{-3}$); t is the time interval (10 min); Q is the volume of the outflow through the soil cores during the time interval t (ml); L is the length of the soil core (5 cm); $\Delta H$ is the height difference of the hydraulic head (10 cm); A is the cross-sectional area of the cylinder core ($20 \ cm^2$).

Above all, soil moisture content (SMC; volumetric) of all investigated sites was measured by time domain reflectometry (TDR) (TRIME-PICO-IPH, TDR, IMKO Inc., Ettlingen, Germany) from 0–80 cm soil depth at 10 cm intervals from June 20 to July 20, 2014. The TDR was calibrated in the local alpine region in advance, and the determination accuracy was ±3%. Soil moisture was measured three times for each layer of each site. There were no rainfall events recorded within 2 days before the data collection, and each measurement of all sites were finished within one day.

## 2.3 Statistical analysis

Data in this study were presented as mean±SD (standard deviation). Comparison analysis was performed using SPSS 19.0. A one-way analysis of variance (ANOVA), followed by least significant difference (LSD) method was used to test the

differences between average values of all parameters at each degradation degree. Pearson correlation coefficient was employed to determine the relationships among vegetation degradation and soil properties.

Redundancy analysis (RDA) was applied to study the relationship between basic soil properties and hydraulic properties by using CANOCO software version 4.5 (Biometris). RDA is a type of constrained ordination method combining multiple regression and principal component analysis (PCA). It aims to represent a multivariate data set (generally a collection of samples with more than two properties) along a reduced number of orthogonal axes, and visualize the data set in a two-dimensional scatter diagram, hence enabling an easier interpretation of the structure of multivariate data and relationships among variables (Borcard et al., 2011). The projections of the arrows onto the axes represent the contribution of corresponding variables to the extracted axes. The cosine of the angle between the arrows reflects the correlation between variables. Monte-Carlo permutation test was used to rank the importance of each explanatory variable, and then the contribution of each variable to the total variance was determined by multiple regression (Leps, 2003).

## 3 Results

### 3.1 Variation in basic soil properties and porosity characteristics under different degrees of degradation

Changes in basic soil properties and porosity characteristics with alpine swamp meadow degradation were obvious (Fig. 3). Statistical analysis showed that SOC and WSA decreased significantly ($p<0.05$; Figs. 3b, c) and BD increased significantly ($p<0.05$; Fig. 3a) with degradation. Soil texture changed remarkably with degradation; sand content increased significantly ($p<0.05$; Fig. 3f) while significant decreases were observed in silt and clay content ($p<0.05$; Figs. 3d, e). The majority of all soil samples were classified as loam and sand (Fig. 4). Half of the LD samples were classified as loam, while the vast majority of MD (17 of 24 samples) and SD (22 of 24 samples) samples were classified as sand. Compared with LD, SOC, WSA and silt and clay content of MD decreased by 17.9%, 15.7%, 5.1% and 23.1%, respectively, and those of SD decreased by 61.5%, 32.8%, 44.0% and 75.8%, respectively. BD and sand content of MD increased by 2.3% and 2.9%, respectively, and those of SD increased by 7.2% and 19.6%, respectively, when compared with LD.

Soil porosity altered drastically with degradation (Figs. 3g, h). CP decreased consistently with increasing degree of degradation. When compared with LD, mean CP values of all depths decreased by 5.5% and 13.6% for MD and SD, respectively. The mean value of NCP decreased from LD to MD by 6.6% while it increased from MD to SD by 4.4%, following the order of LD>SD>MD.

All properties differed most distinctly in surface (0–10 cm) and subsurface layers (10–20 cm) among different degradation degrees. The differences gradually diminished with increasing soil depth despite some exceptions (e.g., 40–50 cm for clay and 70–80 cm for silt). Almost all basic soil properties showed strong depth dependence. For each degradation degree, BD

and sand content showed an increasing trend while SOC, WSA and silt and clay content decreased consistently with depth from 0–80 cm. CP of MD experienced parabolic change with the highest values in the 20–30 cm layer. NCP was an exception, showing decreases from 0–40 cm while increasing slightly from 40–80 cm. For each property, the slope of the vertical variations decreased with degradation.

## 3.2 Changes in SMC, FC and Ks with degradation

SMC in the profile decreased consistently with degradation for all soil layers (Fig. 5). Compared with LD, the mean SMC (0–80 cm layer) of MD and SD decreased by 21.8% and 33.5%, respectively, and SMC declined more from LD to MD than from MD to SD. SMC of different degradation degrees always showed an increasing trend with depth. For MD and SD, SMC increased consistently with depth, while for LD, SMC showed no clear trend in both 0–40 cm and 40–80 cm layers, increasing sharply at 40 cm depth.

Changes in FC and Ks associated with alpine swamp meadow degradation are displayed in Fig. 6. Both of these properties responded quickly to degradation and showed notable vertical distribution. Mean values of FC decreased consistently with degradation in 0–30 cm layers but varied irregularly below 30 cm (Fig. 6a). Unlike FC, Ks decreased from LD to MD and then increased from MD to SD (i.e., LD>SD>MD) except for layers 40–50 and 70–80 cm (Fig. 6b). It was also evident that Ks values were more variable in the upper soil layers. FC of all degradation degrees decreased consistently with depth, and the slope of the decreasing trend decreased with degradation, while Ks decreased in the 0–40 cm layers and then increased in the 40–80 cm layers, reaching the lowest values at 40 cm. Similar patterns of change and vertical distribution were observed for NCP (see section 3.1).

ANOVA showed that SMC of LD was significantly higher ($p<0.05$) than that of MD in all soil layers except for the 20–30 cm layer, and SMC of MD was significantly higher than that of SD in the 10–80 cm layers. In contrast to SMC, significant differences among the three degrees of degradation only existed in the 0–20 cm layers for FC, and the 0–10 cm layer for Ks. These statistical analyses indicated that alpine meadow degradation did not have significant impacts on soil hydraulic properties in layers deeper than 20 cm.

## 3.3 Influencing factors of degradation on soil hydraulic properties

Alpine swamp meadow degradation directly leads to deterioration of basic soil properties, and thus soil hydraulic properties are influenced. Pearson correlation analysis (Table 2) showed the effects of vegetation characteristics on soil properties. Generally, SOC, WSA, CP and clay and silt content were positively correlated with VC, MA and MB, while BD and sand content were negatively correlated with vegetation characteristics. The correlation between NCP and vegetation characteristics was not significant because of its inconsistent change pattern with degradation. Compared with those in the 40–80 cm layers, soil properties in the 0–40 cm layers were all significantly ($p<0.05$ or $0.01$) correlated with vegetation

characteristics, indicating that the effects of vegetation degradation on soil properties were mostly confined to the upper soil layers.

According to the statistical analysis in section 3.2, data of samples in the 0–10 and 10–20 cm layers were selected to analyse the relationships between basic soil properties and hydraulic properties associated with degradation. Figure 7 illustrates the relationships between soil basic and hydraulic properties ascertained by the RDA. FC was positively correlated with CP, WSA, SOC and silt and clay content, but was negatively correlated with BD and sand content. NCP had no impact on FC, but served as the only factor that determined Ks. FC and Ks are independent of each other, which can be further supported by Pearson correlation analysis (Table 3). The two axes explained 60.2% and 29.0% of the total variance in FC and Ks, respectively.

Additionally, all the samples could be divided into two groups: one included all the samples from LD and two samples from SD, while the other included all the samples from MD and four samples from SD. The group including LD samples showed a close relationship with all the soil properties except BD and sand, while the second group mainly including MD and SD showed the opposite.

Soil basic properties were treated as explanatory variables to explain the total variance of Ks and FC. A Monte-Carlo permutation test was first used to rank the importance of each explanatory variable, and then the relative contributions of each variable to the total variance of Ks and FC were determined by multiple regression. The results showed (Table 4) that CP was the dominant factor for FC, explaining 91.9% of the variance in FC. NCP was the dominant factor for Ks, explaining 97.0% of the variance in Ks. As these properties explain large proportions of the variance in FC and Ks, the relative influence of other soil properties can be dismissed.

## 4 Discussion

### 4.1 Effects of alpine swamp meadow degradation on SMC and basic soil properties

SMC is a comprehensive indicator of soil quality and can directly reflect soil water holding capacity (Palese et al., 2014; Zeng et al., 2013). This study showed that SMC decreased consistently from LD to SD, responding swiftly to degradation. Unlike soil properties, significant differences in SMC among the three degradation degrees were found at all soil layers. Similar changing patterns with vegetation degradation in alpine regions were observed by Zeng et al. (2013), Wang et al. (2007) and Li et al. (2012). In fact, decreases in vegetation coverage and SOC, and increase in sand content will negatively impact on soil water retention (Strudley et al., 2008; Yi et al., 2012), leading to loss of SMC with degradation. Moreover, because of the intensified root uptake and soil evaporation in summer, SMC of all degradation degrees in the 0–30 cm layers were lower than those in deeper layers.

Changes in basic soil properties, such as increases in BD and sand content and decreases in SOC, WSA and silt and clay content with degradation (Figs. 2a–f) align closely with the hypothesized results and are in agreement with much of the literature (Gao et al., 2010; Wang et al., 2007; Wei et al., 2010; Zeng et al., 2013). Along the degradation gradient, trends in these basic properties are almost uniform regardless of soil types and vegetation traits (Guo et al., 2013; Zeng et al., 2013).

On the contrary, basic soil properties will improve consistently during restoration processes (Li et al., 2006; Wu et al., 2010). Moreover, changes in basic soil properties with degradation were mostly confined to the upper layers (0–40 cm), and the contrasts were most remarkable in the surface layer (0–10 cm). Similar change patterns with soil depth have also been confirmed in many other studies (Fu et al., 2015; Hallema et al., 2015; Wang et al., 2007; Zeng et al., 2013). Correlation analysis showed that soil properties, especially in the upper layers (0–40 cm), were closely associated with vegetation

characteristics (Table 2). Vegetation factors are indispensable to the stability of soil status, such as the formation of SOC, porosity and structure. In fact, root activity and litter fall input decrease significantly or disappear as degradation degree increases, thus the decomposition process and organic matter accumulation in soil are hindered with degradation. Depletion of SOC greatly alters the soil micro-environment and might trigger a series of changes in soil physical, chemical and biological processes (Nelson et al., 1996). For instance, it has been confirmed that clay and silt contents are largely

dependent on the release of organic acid from soil organic matter, which can corrode coarse minerals and transfer large grains into fine particles (Fan et al., 2015). Organic matter can also act as "glue" in soil aggregates formation and determine water stability (Lipiec et al., 2009). Therefore, a decrease in SOC will strongly influence soil structure, and thus brings about overall changes in soil physical and chemical properties. Furthermore, the absence of plant coverage and root grasp will cause topsoil to become vulnerable to wind, raindrops, surface flow and compaction, directly resulting in soil erosion and

degradation, and the particle distribution of soil samples in the soil texture triangle (Fig. 4) clearly shows the sandification trend with increasing degradation. Insignificant correlations between vegetation characteristics and soil properties of the deep layers (40–80 cm; Table 2) were in accordance with the fact that no significant differences in soil basic properties were found among degradation degrees in the deep layers.

## 4.2 Influencing factors of alpine swamp meadow degradation on soil hydraulic properties

With increasing degradation degree, our results demonstrated that FC decreased consistently (Fig. 6a), which was in accordance with some previous studies conducted in other alpine ecosystem types (Xiong et al., 2011; Yi et al., 2012). However, other studies reported different change patterns of FC. Li et al. (2012) and Wei et al. (2010) found that FC first increased but then decreased with alpine meadow degradation. This might be caused by different degradation classification. Li et al. (2012) and Wei et al. (2010) classified the sampling plots into four and five degradation degrees according to

succession stages, respectively, and the initial degree of the sampling plots in both studies was non-degraded. However, the highest values of FC in both studies corresponded to the light degree, and FC of the most severely degraded sites were much lower than those of non-degraded sites. These traits were in line with our results. Wei et al. (2010) noted that CP changed

similarly with degradation; i.e., CP was positively correlated with FC, which was consistent with the definition of FC (Ottoni Filho et al., 2014).

Ks decreased initially and then increased with degradation (Fig. 6b), and similar "high–low–high" trends in Ks were also observed in some studies conducted for alpine meadow, although the amplitude of change was different (Wang et al., 2010; Wei et al., 2010). However, Zeng et al. (2013) reported a decreasing trend while Wang et al. (2007) reported an increasing trend of Ks. These discrepancies might be attributed to the difference in soil and vegetation factors in different regions and also the degradation classification. In the study conducted by Zeng et al. (2013), the root density consistently decreased by an average of 97% from light to extreme degree, which indicated a substantial decrease in soil macro-pores and thus resulted in lower values of Ks; in the study conducted by Wang et al. (2007), the gravel (size >2 mm) content increased by nearly 100-fold, which will greatly increase soil infiltration.

Soil pores are empty space active in soil water storage, retention and movement, therefore soil porosity is closely related to soil hydraulic properties (Lipiec et al., 2006). Increases in BD indicate a reduction in soil total porosity (TP) since TP is generally calculated using the following equation: $TP = 1 - BD/2.65$ (Li et al., 2006; Price et al., 2010). TP can be divided into CP (pore size <0.1 mm) and NCP (pore size >0.1 mm). Water that fills capillary pores can be suspended by the capillary effect, making CP vital for soil water retention. However, in non-capillary pores soil water can move freely by gravity making NCP critical for soil water infiltration and transmission. Generally, soil pores with pore size larger than 75 µm are defined as macro-pores, and non-capillary pores belong to the category of macro-pores (Gao et al., 2015; Pagliai et al., 2002). Soil porosity mainly depends on soil texture and aggregates (i.e., the finer the texture of the soil, the smaller the pore size). Applying this logic, increasing degradation would increase sand content and decrease WSA, and thus lead to a decrease in CP (Fu et al., 2015; Lipiec et al., 2006). Moreover, a positive correlation between CP and SOC was detected in many studies (Gao et al., 2015; Price et al., 2010; Yu et al., 2015), reflecting that CP can be an indicator of soil quality. By definition, FC is the maximum water content held in soils when excess water has drained away and the downward flux is negligible (Ottoni Filho et al., 2014). Therefore, FC is essentially dependent on the capillary effect; decreased FC is the result of a decrease in CP.

Unlike CP, changes in NCP in this study were more complex; NCP first decreased from LD to MD and then increased from MD to SD (i.e., MD<SD<LD; Fig. 3h). It is widely accepted that soil macropores are closely related to root penetration and activities of soil fauna (Kuncoro et al., 2014; Zeng et al., 2013). NCP measured in the rhizosphere (0–20 cm layer) decreased significantly as root penetration weakened with degradation. In contrast, increases in sand content will lead to an increase in the size of soil pores. Hence, the slight but observed increase in NCP from MD to SD. However, the effect is not equivalent with root penetration resulting in macro-pores; i.e., the contribution of increasing sand content to NCP could not offset the diminishing effects of vegetation on soil porosity, and hence NCP of SD was higher than that of MD but still lower than that

of LD. Ks determines soil water movement and is largely dominated by NCP (Tables 3 and 4), so NCP changed in accordance with Ks.

In summary, our study revealed that the well-identified relationships between soil porosity and hydraulic properties are applicable in alpine swamp meadow. Hence, compared with soil porosity, the contributions of other properties to the variance of FC and Ks were outweighed (Table 4). In addition, FC was positively correlated with SOC, WSA and silt and clay content ($p<0.05$), and negatively correlated with BD and sand content ($p<0.05$; Table 3); these correlations were consistent with studies in other regions (Głąb et al., 2014; Price et al., 2010; Wei et al., 2010). Because of the inconsistent changing pattern, Ks only positively correlated with NCP (Table 3). In fact, arguments about the impact of soil properties on Ks and its changes are still under debate (Fu et al., 2015; Jarvis et al., 2013). Hence, further investigations about variations of Ks are needed.

### 4.3 Implications and uncertainties of this study

Our results show a clear distinction of basic soil and hydraulic properties among different alpine meadow degradation degrees. Considering the important roles that FC and Ks play in soil water retention and infiltration, it can be concluded that key hydraulic processes and functions in soil such as water holding capacity, transmission and runoff generation mechanisms might differ significantly with alpine swamp meadow degradation. For example, high Ks seen in topsoil can form preferential flow and avoid infiltration excess runoff (Fu et al., 2015; Lipiecet al., 2006). In this study, soils of LD had relatively high Ks and FC, indicating the robustness of soil water retention. For MD, Ks values were reduced significantly; low Ks might act as a barrier to vertical water flow reducing its capacity to intercept rainfall.

Furthermore, the results showed that the effects of degradation mainly manifest in the upper soil layers. There are only a few influences of degradation in deep soil layers. Moreover, the rhizosphere lies at the interface between the atmosphere and the ground surface and directly accepts precipitation, recharges deep soil layers and supplies water to plant growth (Li et al., 2012; Wu et al., 2014). In this sense, the rhizosphere is of great hydrological importance to alpine ecosystems, and changes in soil hydraulic properties of this layer could greatly alter the soil hydraulic processes in local regions.

The hydrological effects of large-scale alpine meadow degradation are noticeable and serious in the Tibetan Plateau (Jin et al., 2015; Wang et al., 2012). For hydrological modelling, accurate parameter acquisition is necessary for simulation accuracy (Vereecken et al., 2015). Our results indicate that hydraulic properties will be altered significantly both vertically and spatially with degradation. Therefore, to improve the performance of hydrological modelling, differences in soil hydraulic properties under different degradation degrees should be seriously considered (Jin et al., 2015).

Despite the fact that this study revealed the effect of alpine meadow degradation on soil hydraulic properties, some uncertainties still exist. First, it should be noted that degradation is a non-linear and consecutive process while in practice

people have to divide it into a limited number of degrees according to some criteria. In section 3.2, we have pointed out that the change patterns of FC varied with degradation classification, and actually our classification was relatively rough. Therefore, to obtain robust conclusions about alpine grassland degradation on soil hydraulic properties, more alpine grassland plots should be established and more degradation degrees should be classified in future investigations. Terrestrial ecosystem degradation is essentially a positive feedback loop composed of vegetation retrogressive succession and soil deterioration (King and Hobbs, 2006). Thus, understanding the effect of vegetation degradation on soil hydraulic properties is somewhat insufficient, thereby the interactions between vegetation and soil hydrology should be addressed in further studies.

## 5 Conclusions

Because of global change and anthropogenic disturbances, alpine swamp meadow on the eastern Tibetan Plateau is undergoing severe degradation. Based on nine plots representing alpine swamp meadow of different degradation degrees, this study mainly investigated the changes in soil hydraulic properties with alpine swamp meadow degradation, and analysed the influencing mechanism of grassland degradation on FC and saturated hydraulic conductivity (Ks). In summary, with increasing degradation degree, SMC and FC decreased consistently from LD to SD, while Ks decreased from LD to MD and then increased from MD to SD (i.e., LD>SD>MD). BD, SOC, WSA, soil texture and porosity were also substantially altered. Significant differences in both soil basic and hydraulic properties between different degradation degrees usually exist in the 0–20 cm layer, indicating that the effect of degradation was mostly concentrated in the upper soil layers and the rhizosphere. FC was positively correlated with CP, WSA, SOC and silt and clay content, but was negatively correlated with BD and sand content; Ks was only positively correlated with NCP.

Changes in FC and Ks are mainly controlled by soil porosity during the degradation process. CP and NCP are dominant factors, which explained 91.1% and 97.3% of the variance of FC and Ks, respectively. Root activities attenuate with degradation and directly lead to a decrease in NCP, while the contribution of sand particles to NCP is important for MD and SD when vegetation diminishes or disappears. The combined effect of disappearing root activities and increased sand content is responsible for the inconsistent changes in NCP and Ks during the degradation processes. Our findings provide a more comprehensive understanding of the soil hydrological effects of vegetation degradation. Given the importance of parameterization for hydrological models, water flow simulations in the Tibetan Plateau and similar regions should consider variations in soil hydraulic properties of different degraded alpine swamp meadow.

## Acknowledgments

This study was supported by the National Natural Science Foundation of China (41671107 & 41301092) and the Youth Innovation Promotion Association of the Chinese Academy of Sciences (2016049). We thank Professor Zhang Yu, Dr.

Wang Shaoying and Zhao Wanglong for their help with field work. We are also in debt to Dr. Gao Xiaofei for his assistance in soil treatment and analysis. In addition, we appreciate the anonymous reviewers' valuable suggestions to improve the paper.

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

**Table 1. Vegetation characteristics of investigated sites in this study.**

| Degradation Degree | VC (mean±SD*, %) | Number of species | Dominant species | |
|---|---|---|---|---|
| LD | 80.5±4.9 | 18-25 | *Kobresiatibetica, Stipaaliena* | *Kobresiahumilis,* |
| MD | 59.7±4.5 | 15-20 | *Kobresiapygmaea, Carextristachya* | *Agropyroncristatum,* |
| SD | 13.7±8.6 | 5-12 | *Kobresiarobusta, Potentillabifurca* | *Leymuschinensis* |

Note: *, standard deviation

**Table 2. Pearson correlation coefficients between vegetation characteristics and soil properties in 0–40 cm and 40–80 cm layers**

| Layers | Properties | BD | SOC | WSA | Sand | Silt | Clay | CP | NCP |
|---|---|---|---|---|---|---|---|---|---|
| 0-40 cm | VC | -0.710** | 0.769** | 0.747** | 0.533* | 0.472* | -0.491* | 0.829** | 0.155 |
| | MA | -0.811** | 0.899** | 0.902** | 0.838** | 0.698** | -0.735** | 0.808** | 0.345 |
| | MB | -0.635** | 0.860** | 0.800** | 0.672** | 0.646** | -0.662** | 0.615** | 0.028 |
| 40-80 cm | VC | -0.187 | 0.658** | 0.586** | 0.249 | 0.420 | -0.405 | 0.321 | -0.407 |
| | MB | -0.487* | 0.461* | 0.464* | 0.365 | 0.507* | -0.502* | 0.544* | -0.030 |
| | MA | -0.352 | 0.474* | 0.461* | 0.369 | 0.694** | -0.661** | 0.412 | 0.010 |

Note: *, significant at 0.05 level; **, significant at 0.01 level (2-tailed test); n=18.

**Table 3. Pearson correlation coefficients between Ks, FC and soil properties of soil in layers above 20 cm depth.**

| Properties | BD | SOC | WSA | Sand | Silt | Clay | CP | NCP |
|---|---|---|---|---|---|---|---|---|
| Ks | -0.447 | -0.239 | -0.246 | -0.381 | 0.366 | 0.391 | 0.172 | 0.896** |
| FC | -0.912** | 0.867** | 0.875** | -0.803** | 0.786** | 0.760** | 0.918** | 0.361 |

Note: **, significant at 0.01 level (2-tailed test); $n$=18.

**Table 4. Total variance of FC and Ks explained by basic soil properties**

| Ranking | FC | | | Ks | | |
|---|---|---|---|---|---|---|
| | Properties | % of Variance | Cumulative% | Properties | % of Variance | Cumulative% |
| 1 | CP | 91.1 | 91.1 | NCP | 97.3 | 97.3 |
| 2 | WSA | 7.5 | 98.6 | BD | 1.8 | 99.1 |
| 3 | NCP | 0.7 | 99.3 | WSA | 0.5 | 99.6 |
| 4 | Silt | 0.5 | 99.8 | CP | 0.2 | 99.8 |
| 5 | BD | 0.2 | 100.0 | Clay | 0.1 | 99.9 |
| 6 | SOC | 0.0 | 100.0 | Silt | 0.1 | 100.0 |
| 7 | Clay | 0.0 | 100.0 | Sand | 0.0 | 100.0 |
| 8 | Sand | 0.0 | 100.0 | SOC | 0.0 | 100.0 |

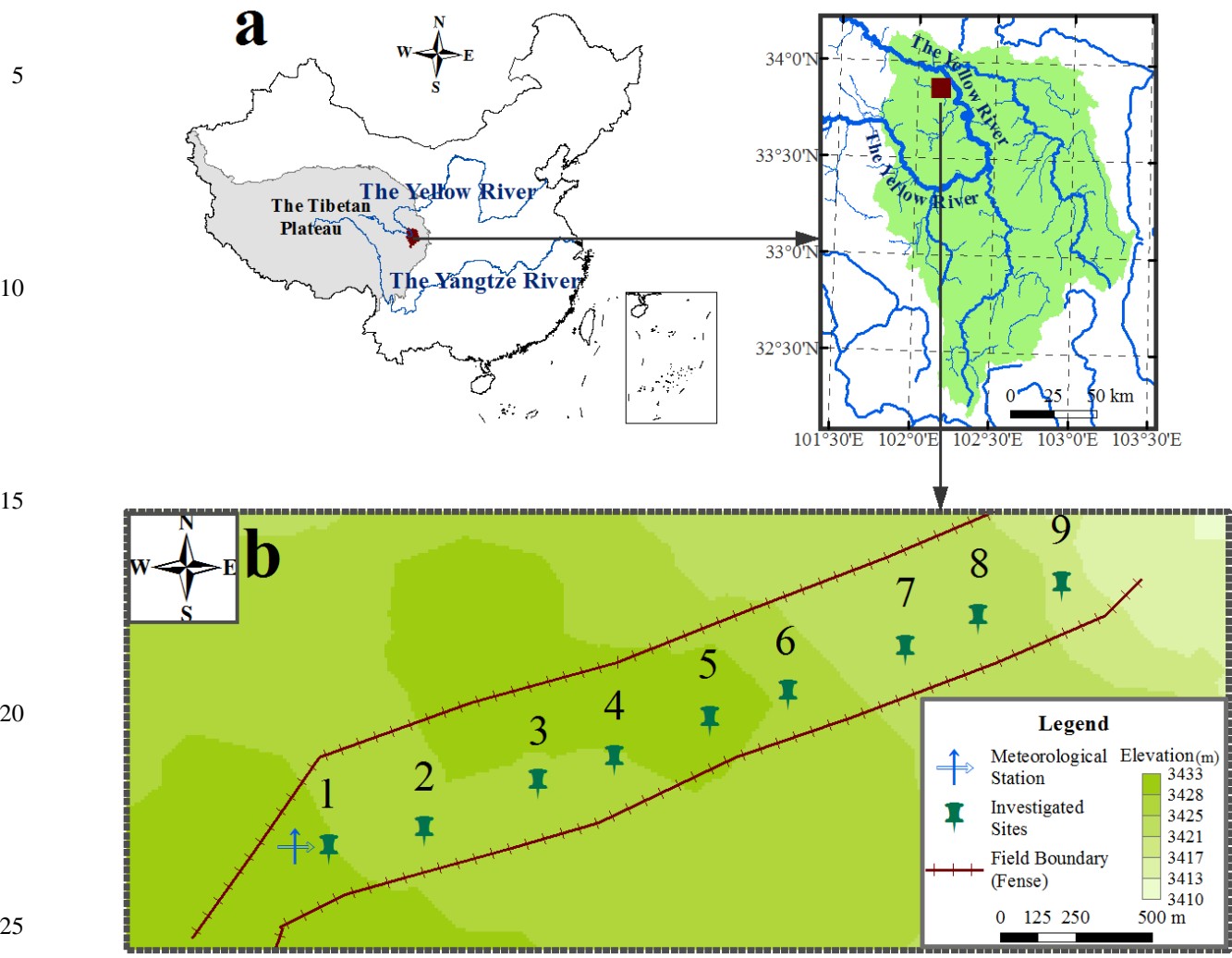

**Figure 1.** Location of the study area and investigated sites: a) location of the experimental field in the Zoige Wetland in the east of the Tibetan Plateau, China; b) distribution of investigated sites within the experimental fields.

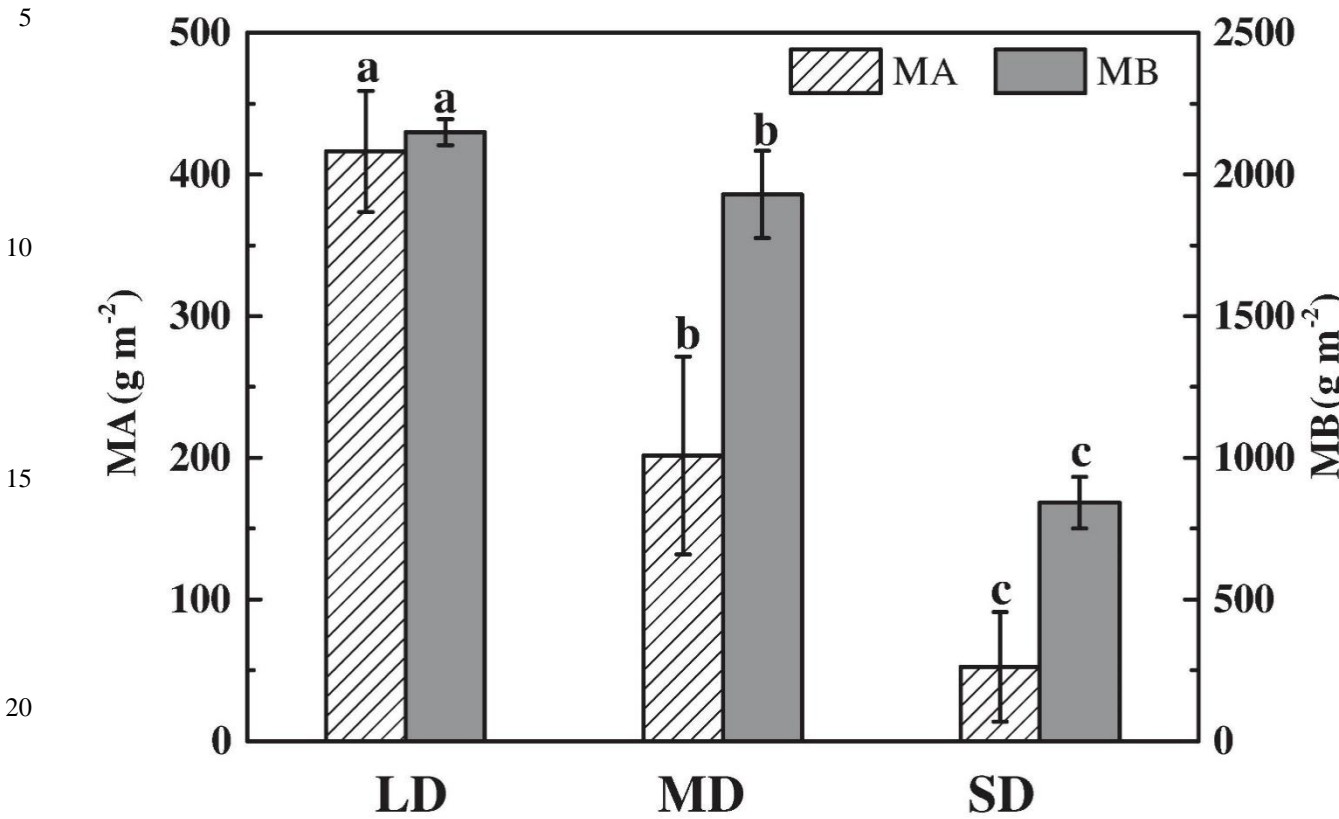

**Figure 2.** MA and MB of different degradation degrees. The error bars denote the standard deviation of the 3 sites of the same degradation
degree. Different letters above the bars denote significant differences ($p<0.05$) between different degradation degrees. LD: lightly
degraded, MD: moderately degraded, SD: severely degraded.

30

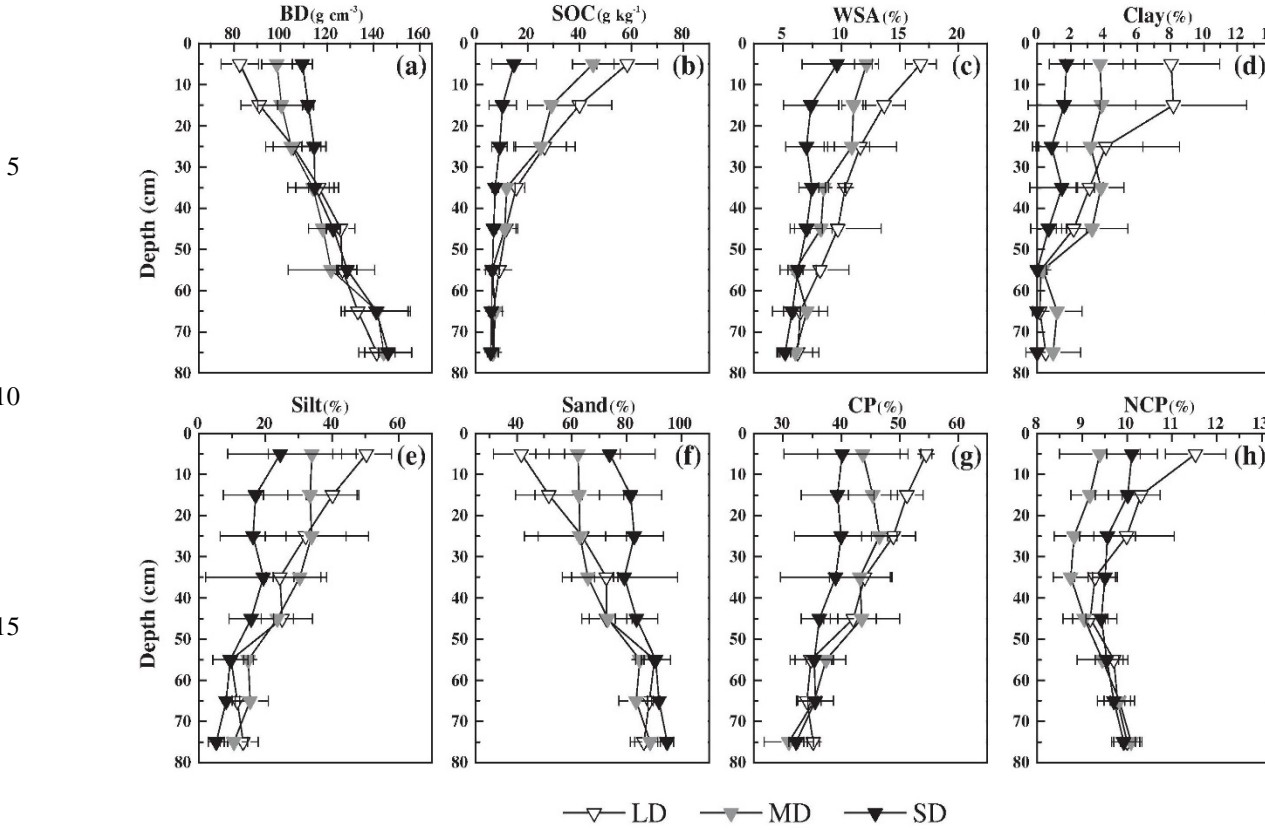

**Figure 3.** Basic soil properties of different degradation degrees. The error bars denote the standard deviation of the three sites of the same degradation degree. LD: lightly degraded, MD: moderately degraded, SD: severely degraded.

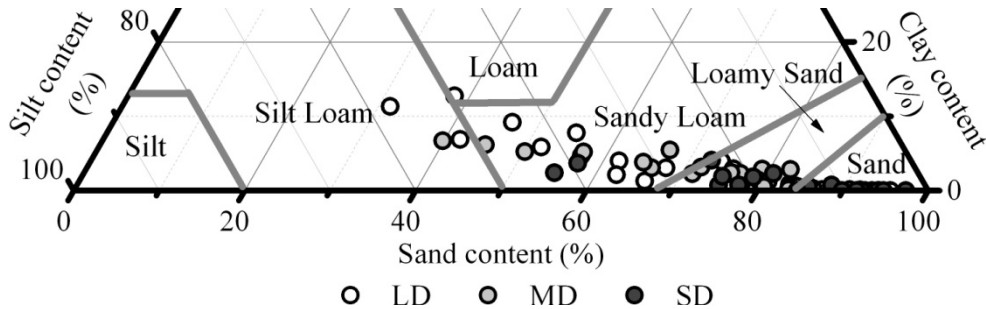

**Figure 4.** Particle size distributions of lightly degraded (LD), moderately degraded (MD), and severely degraded (SD) soil samples. Textural classes corresponding to particle size distributions observed in these soils are bounded by grey bold lines (e.g., loam, silt).

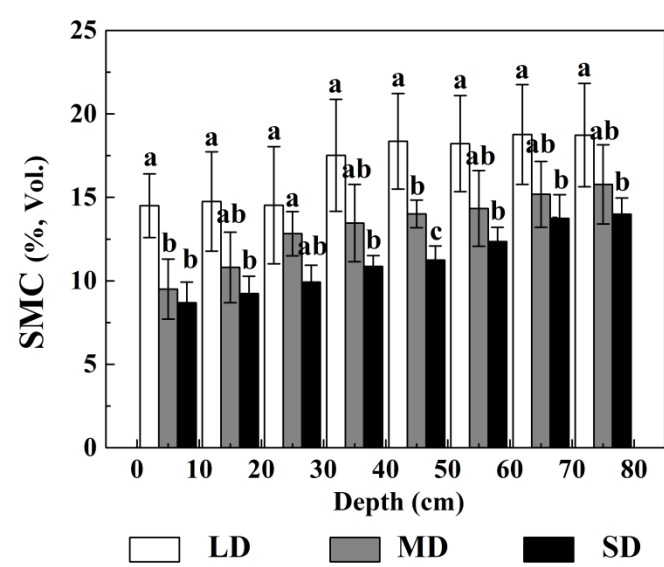

**Figure 5.** Soil moisture content (SMC) of different degradation degrees. Error bars denote the standard deviation of the 3 sites of the same degradation degree. Bars with the same letter indicate that no significant differences ($p<0.05$) exist between corresponding degradation degrees. LD: lightly degraded, MD: moderately degraded, SD: severely degraded.

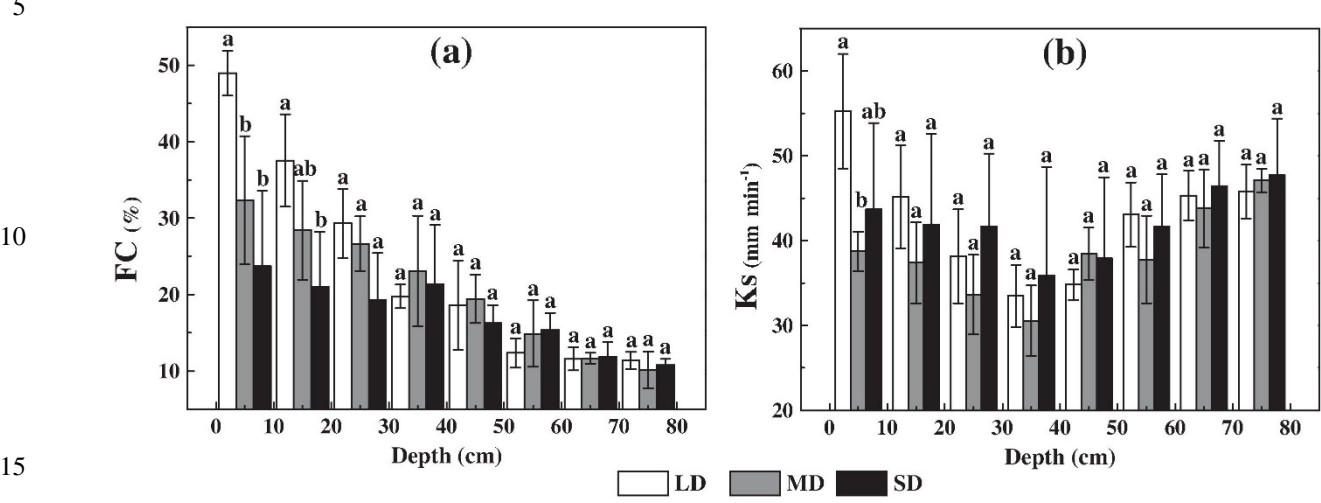

**Figure 6.** Differences in field capacity (FC) and saturated hydraulic conductivity (Ks) with degradation degree. Error bars denote the standard deviation of the 3 sites of the same degradation degree. Bars with the same letter indicate that no significant differences ($p<0.05$) exist between corresponding degradation degrees. LD: lightly degraded, MD: moderately degraded, SD: severely degraded.

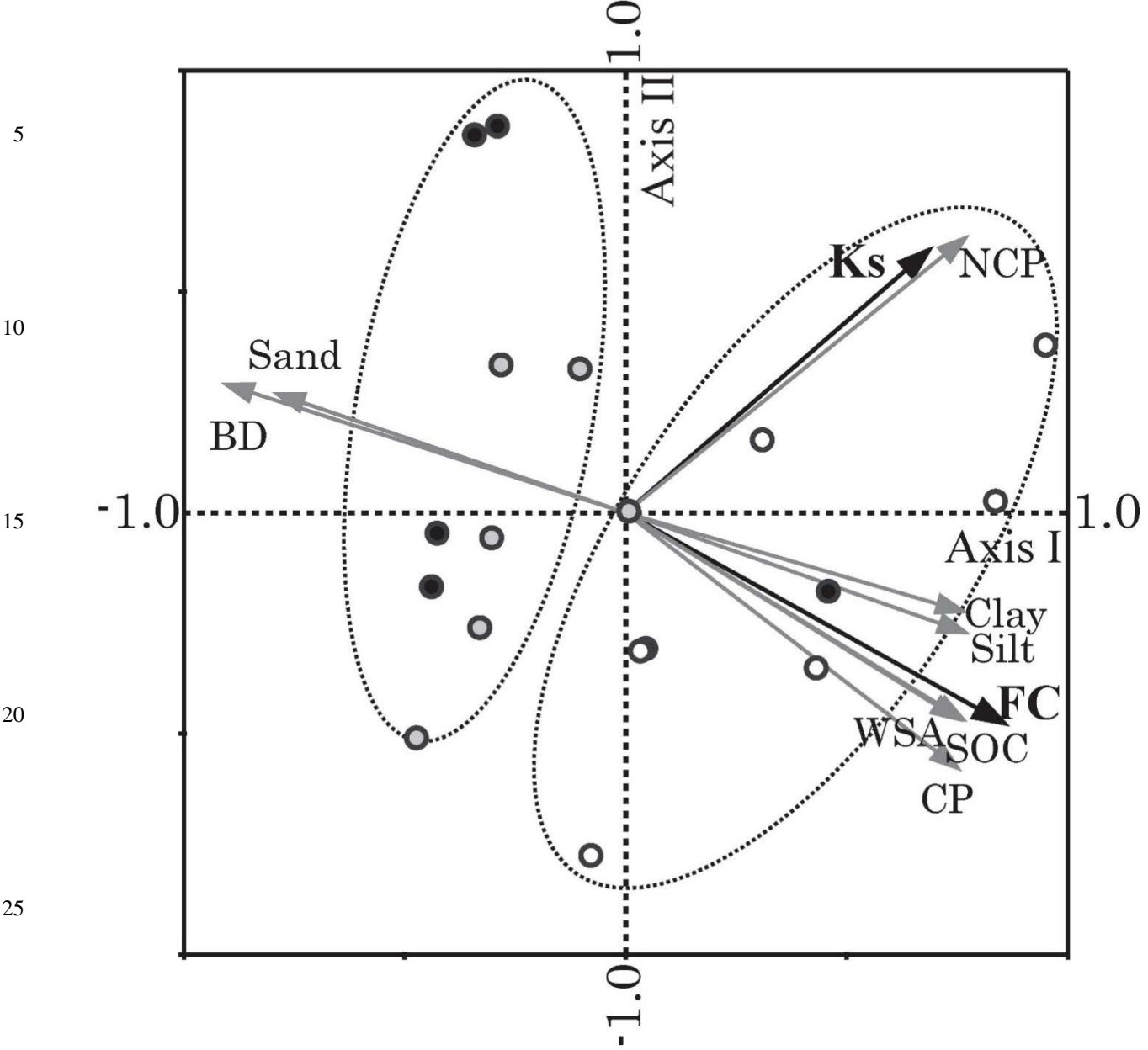

**Figure 7.** Redundancy analysis of soil hydraulic properties and basic properties under different degradation stages. Symbols '○', '●' and '●' denote soil samples from lightly degraded (LD), moderately degraded (MD) and severely degraded (SD), respectively. The two axes represent the principal component (PC) extracted from the explanatory variables (basic soil properties). The first ordination axis (axis I, horizontal) mainly reflected the influence of bulk density (BD), soil organic carbon (SOC) and soil texture and the second axis (axis II, vertical) mainly reflected the influence of capillary porosity (CP) and non-capillary porosity (NCP). FC: field capacity, Ks: saturated hydraulic conductivity, WSA: water-stable aggregates.