# Peer review of "Variation of soil hydraulic properties with alpine grassland degradation in the Eastern Tibetan Plateau"

_Hydrology and Earth System Sciences, 2016_

## Referee Comment (RC1) · Anonymous Referee #1 · 31 Aug 2016

This study investigated the variation of soil hydraulic properties (soil field capacity and saturated hydraulic conductivity) in different degraded alpine grassland fields in the Tibetan Plateau. The basic soil properties and hydraulic properties were compared, and the dominant soil factors of soil hydraulic properties were identified. This study is important for the hydrological modeling and ecosystem management in alpine mountainous regions. My main concern is that this study seems like a local study and the novelty is somewhat lower than the standard of HESS. The current version needs major revision.

1. In the Introduction section, the authors should substantially review the relevant studies in alpine mountainous regions, not just Tibetan Plateau of China. The main findings, discrepancies and weaknesses of previous studies and the motivations of this study should be addressed in detail.

[Figure]

2. The authors indicated that "large discrepancies still exist in the obtained conclusions and knowledge gap remains". However, in the Discussion section, the authors pointed out several times that most results of this study were consistent with previous studies, (such as P.7, Line 25, "in agreement with", P.8, Line 19, "is consistent with", P.8, Line 31, "The similar", P.9, Line 10, "consistent with"). What are the new and different findings of this study with respect to those in Tibetan Plateau of China, and more imporant other alpine mountainous regions in the world. What is the reason and mechanism for the differences? The authors should substantially address them and improve the highlights.

3. The authors only investigated the effects of soil properties on hydraulic properties. I think the role of vegetation characteristics including roots should be included in the analysis. The degradation changed both vegetation and soil characteristics to affect soil hydraulic properties.

4. If the authors also measured soil moisture, it is necessary to compare soil water content among different degraded alpine grassland fields.

In summary, the novelty of this study should be improved, and the comparisons and discussions with studies in other alpine mountainous regions in the world should be addressed.

---

## Author Comment (AC1) · 2 Oct 2016

Comment 1 In the Introduction section, the authors should substantially review the relevant studies in alpine mountainous regions, not just Tibetan Plateau of China. The main findings, discrepancies and weaknesses of previous studies and the motivations of this study should be addressed in detail. Reply: We appreciate and agree with the reviewer very much for the constructive comment. Indeed, we paid most attention to the Tibetan Plateau of China in the introduction section which was not a substantial review for relevant studies in alpine mountainous regions. In the revised manuscript, relevant studies in other similar alpine mountainous regions such as the south of Tibetan Plateau in Nepal, the Alps mountainous area in Europe, the high land in North China etc. were added in the introduction section. At the same time, we revised the descriptions of main findings, discrepancies and weaknesses of previous studies of

this study carefully in the revised manuscript. And hence the motivations of this study were addressed in detail. For specific revisions and changes, please see the revised manuscript appended.

Comment 2 The authors indicated that "large discrepancies still exist in the obtained conclusions and knowledge gap remains". However, in the Discussion section, the authors pointed out several times that most results of this study were consistent with previous studies, (such as P.7, Line 25, "in agreement with", P.8, Line 19, "is consistent with", P.8, Line 31, "The similar", P.9, Line 10, "consistent with"). What are the new and different findings of this study with respect to those in Tibetan Plateau of China, and more important other alpine mountainous regions in the world. What is the reason and mechanism for the differences? The authors should substantially address them and improve the highlights. Reply: We totally agree with the reviewer's comment. As the reviewer pointed out, our description in the introduction and discussion was a little bit contradictive and misleading. For the first "in agreement with" in P.7, Line 25, we would like to give a further explanation that here we mainly talked about the variation of basic soil properties including bulk density, soil organic carbon and etc., not soil hydraulic properties. As for other points the reviewer issued, we substantially compared our findings to previous studies in the revised manuscript and discussed the consistent and inconsistent results. Actually, there were several different findings comparing to existed studies. So the discussion section was carefully revised. We also avoided using suchlike misleading words in the discussion section. Thereafter we pointed out the reasons and mechanism for the different results and improved the highlights. For specific revisions and changes, please see the revised manuscript appended.

Comment 3 The authors only investigated the effects of soil properties on hydraulic properties. I think the role of vegetation characteristics including roots should be included in the analysis. The degradation changed both vegetation and soil characteristics to affect soil hydraulic properties. Reply: We thanked for the informative suggestion. In site selection, we fully considered and investigated the vegetation characteristics of each degradation degrees, like coverage, biomass (both above and underground), species number and etc. We found soil organic carbon, bulk density and soil texture, especially those of the top soil, responded swiftly to the changes of vegetation characteristics and changed consistently with degradation degrees, so changes in soil properties partly contain the changes in vegetation characteristics. As the request of the reviewer, we addressed more about the effect of vegetation characteristics, especially the root activity on soil hydraulic properties carefully and tried to explained the mechanism in detail. For specific revisions and changes, please see the revised manuscript appended.

Comment 4 If the authors also measured soil moisture, it is necessary to compare soil water content among different degraded alpine grassland fields. Reply: Thanks a lot for the reviewers suggestion and reminding. Actually, we have measured soil moisture content of all investigated sites in the summer of 2014 from June 20th to July 20th. So as suggested by the reviewer, the comparison of soil moisture content among different degraded alpine grassland was added in the paper. For specific revisions and changes, please see the revised manuscript appended.

Please also note the supplement to this comment:
http://www.hydrol-earth-syst-sci-discuss.net/hess-2016-333/hess-2016-333-AC1-supplement.zip

---

## Referee Comment (RC2) · Anonymous Referee #2 · 12 Oct 2016

1. The study analyze the effects of alpine grassland degradation on soil hydraulic properties in Tibetan Plateau. Nine sites representing various degradation degrees were selected, and field and laboratory experiment were applied. The study give the confident results by the abundant data and detailed explanation, and it will contributing to understand the soil hydrological effects of vegetation degradation. However, there are several minor problems need to improve before the manuscript can be accept. 2. The conclusion focus on the effects of CP and NCP on FC and Ks, and it should be improved to express more content of research. 3. At the 2.1 part, the VC, DS, and NS were selected as indicators of degradation, the VC is explained before, while the DS and NS need explains here. 4. "Mean values of NCP decreased from LD to MD by 6.6% while increased from MD to LD by 4.4%, following the order of LD>SD>MD.", the presentation is error. 5. The explanation of letters above the bars in Figure 5 and

Figure 6 need improve, and the name of figure 5 need to change. 6. The axis shows of figure 7 is not clear. 7. Monte-Carlo permutation test used in the manuscript to get the Table 3 needed to explain.

———————————————

---

## Author Comment (AC2) · 20 Oct 2016

Dear editorïijŽ We have received the comments from the second anonymous reviewer on our manuscript entitled "Variation of soil hydraulic properties with alpine grassland degradation in the Eastern Tibetan Plateau" (2016-333). These suggestions and comments are quite explicit, aimed at specific problems and errors. Therefore, according to the following comments, we directly adjust and revise related contents in the manuscript. Thank you very much for your work.

Comment 1 The study analyze the effects of alpine grassland degradation on soil hydraulic properties in Tibetan Plateau. Nine sites representing various degradation degrees were selected, and field and laboratory experiment were applied. The study give the confident results by the abundant data and detailed explanation, and it will

contributing to understand the soil hydrological effects of vegetation degradation. However, there are several minor problems need to improve before the manuscript can be accept. Response: We are really grateful for the reviewer's recognition of our work, and these positive comments on the manuscript are quite encouraging. We tried our best to correct the following problems pointed out by the reviewer and thereafter check the manuscript carefully lest any errors and mistakes.

Comment 2 The conclusion focus on the effects of CP and NCP on FC and Ks, and it should be improved to express more content of research Response: We agreed on the comment very much. As the reviewer suggested, we adjusted the content of conclusion and improved to express more content about the grassland degradation impacts on soil basic properties and soil moisture.

Comment 3 At the 2.1 part, the VC, DS, and NS were selected as indicators of degradation, the VC is explained before, while the DS and NS need explains here. Response: We are sorry for our negligence, the two confusing abbreviations have been replaced by the full name in section 2.1

Comment 4 "Mean values of NCP decreased from LD to MD by 6.6% while increased from MD to LD by 4.4%, following the order of LD>SD>MD.", the presentation is error. Response: Thanks for pointing out the error, it should be "MD to SD", and we have corrected the presentation in the revised manuscript.

Comment 5 The explanation of letters above the bars in Figure 5 and Figure 6 need improve, and the name of figure 5 need to change. Response: According to the reviewer's advice, we have further explained the meaning of letters above the bars in Fig 5 and 6 in the figure caption. The name of Fig 5 was also adjusted.

Comment 6 The axis shows of figure 7 is not clear. Response: We adjusted the position of the axis name, making the axis indicating more clear. We also explained more details about the axis in the revised figure caption.

Comment 7 Monte-Carlo permutation test used in the manuscript to get the Table 3 needed to explain Response: We have introduced the method to get the Table 3 in detail, and the relationship between the contribution of total variance and Monte-Carlo permutation test was also clarified in section 2.3..

Please also note the supplement to this comment:
http://www.hydrol-earth-syst-sci-discuss.net/hess-2016-333/hess-2016-333-AC2-supplement.pdf

**Supplement:**

[revised manuscript text omitted]

---

## Author Response (AR1)

Dear editor,

Thank you very much for giving us the opportunity to revise our manuscript entitled "**Variation of soil hydraulic properties with alpine grassland degradation in the Eastern Tibetan Plateau**" **(2016-333)**, and we also appreciate the two anonymous reviewers very much for their constructive comments and suggestions. Those comments are very helpful for revising our paper and quite enlightening on our research. We have studied these comments carefully and tried our best to make corresponding revision and corrections that are waiting for agreement and approval. The point-by-point response to the reviewer's comments are as following:

**Anonymous Referee 1**

**Comment 1** In the Introduction section, the authors should substantially review the relevant studies in alpine mountainous regions, not just Tibetan Plateau of China. The main findings, discrepancies and weaknesses of previous studies and the motivations of this study should be addressed in detail.

**Response:** *We thank the reviewer for the very constructive comment. Indeed, we pay most attention to the Tibetan Plateau of China due to our negligence and mindset, which will result in limitations and even biases in our understanding of alpine soil hydrology. So as the reviewer's request, relevant studies in other similar alpine mountainous regions such as the south of Tibetan Plateau in the Nepal, the Alps mountainous area, the high land in North China were added in the introduction section.*

*As the request of the reviewer, the knowledge gaps exists in alpine soil hydrology were revised based on the new-added researches in combined with that of Tibetan Plateau of China. And hence the motivations of this study were addressed in detail. For specific revisions and changes, please see the revised manuscript appended.*

**Comment 2** The authors indicated that "large discrepancies still exist in the obtained conclusions and knowledge gap remains". However, in the Discussion section, the authors pointed out several times that most results of this study were consistent with previous studies, (such as P.7, Line 25, "in agreement with", P.8, Line 19, "is consistent with", P.8, Line 31, "The similar", P.9, Line 10, "consistent with"). What are the new and different findings of this study with respect to those in Tibetan Plateau of China, and more important other alpine mountainous regions in the world. What is the reason and mechanism for the differences? The authors should substantially address them and improve the highlights.

**Response:** *We totally agree with the reviewer's comment. As the reviewer point out, our description in the introduction and discussion is contradictive and misleading. We confirm that the first "in agreement with" in P.7, Line 25 refers to the basic soil properties including bulk density, soil organic carbon and etc., actually having nothing to do with soil hydraulic properties. Therefore, this one is an exception and need no correction.*

*As the reviewer suggested, we first avoided using suchlike words in the discussion section, and then substantially compared the findings of this studies and those of the previous studies. Thereafter we point out the reason and mechanism for the differences and improve the highlights. For specific revisions and changes, please see the revised manuscript appended.*

**Comment 3** The authors only investigated the effects of soil properties on hydraulic properties. I think the role of vegetation characteristics including roots should be included in the analysis. The degradation changed both vegetation and soil characteristics to affect soil hydraulic properties.

**Response:** *We are grateful for the informative suggestion. In site selection, we fully considered the vegetation characteristics of each degradation degrees, like coverage, biomass (both above and underground), species number and etc. We found soil organic carbon, bulk density and soil texture, especially those of the top soil, responded swiftly to the changes of vegetation characteristics and changed consistently with degradation degrees, so changes in soil properties partly contain the changes in vegetation characteristics.*

*As the request of the reviewer, we more addressed the effect of vegetation characteristics, especially the root activity on soil hydraulic properties. For specific revisions and changes, please see the revised manuscript appended.*

**Comment 4** If the authors also measured soil moisture, it is necessary to compare soil water content among different degraded alpine grassland fields.

**Response:** *Thanks very much for the reviewers suggestion and reminding. Actually, we have measured soil moisture content of all investigated sites in the summer of 2014 for 10 times, and data collection and analysis were also completed. So as suggested by the reviewer, the comparison of soil moisture content among different degraded alpine grassland was added in the paper. For specific revisions and changes, please see the revised manuscript appended.*

**Anonymous Referee 2**

**Comment 1** The study analyze the effects of alpine grassland degradation on soil hydraulic properties in Tibetan Plateau. Nine sites representing various degradation degrees were selected, and field and laboratory experiment were applied. The study give the confident results by the abundant data and detailed explanation, and it will contributing to understand the soil hydrological effects of vegetation degradation. However, there are several minor problems need to improve before the manuscript can be accept.
**Response:** *We are really grateful for the reviewer's recognition of our work, and these positive comments on the manuscript are quite encouraging. We tried our best to correct the following problems pointed out by the reviewer and thereafter check the manuscript carefully lest any errors and mistakes.*

**Comment 2** The conclusion focus on the effects of CP and NCP on FC and Ks, and it should be improved to express more content of research
**Response:** *We agreed on the comment very much. As the reviewer suggested, we adjusted the content of conclusion and improved to express more content about the grassland degradation impacts on soil basic properties and soil moisture.*

**Comment 3** At the 2.1 part, the VC, DS, and NS were selected as indicators of degradation,

the VC is explained before, while the DS and NS need explains here.

**Response:** *We are sorry for our negligence, the two confusing abbreviations have been replaced by the full name in section 2.1*

**Comment 4** "Mean values of NCP decreased from LD to MD by 6.6% while increased from MD to LD by 4.4%, following the order of LD>SD>MD.", the presentation is error.

**Response:** *Thanks for pointing out the error, it should be "MD to SD", and we have corrected the presentation in the revised manuscript.*

**Comment 5** The explanation of letters above the bars in Figure 5 and Figure 6 need improve, and the name of figure 5 need to change.

**Response:** *According to the reviewer's advice, we have further explained the meaning of letters above the bars in Fig 5 and 6 in the figure caption. The name of Fig 5 was also adjusted.*

**Comment 6** The axis shows of figure 7 is not clear.

**Response:** *We adjusted the position of the axis name, making the axis indicating more clear. We also explained more details about the axis in the revised figure caption.*

**Comment 7** Monte-Carlo permutation test used in the manuscript to get the Table 3 needed to explain

**Response:** *We have introduced the method to get the Table 3 in detail, and the relationship between the contribution of total variance and Monte-Carlo permutation test was also clarified in section 2.3.*

[revised manuscript text omitted]

---

## Author Response (AR2)

Dear editor,

We have studied the valuable comments from you and the reviewers carefully. Generally, the greatest concern of the first reviewer is the novelty of this manuscript. He/she thought we didn't deep review the results of previous studies to extract the weakness and subsequently the motivations of this study (#comment 1). After several discussions among authors, we still believe that this manuscript has relative strong novelty but we indeed failed to present it well in the former versions of manuscripts. Therefore, during this round revision, we mainly focus on the introduction section and reviewed more references (details could be found in the revised manuscript) to substantially introduce our original motivation of this research. We also tried our best to address the other comments from reviewers.

Actually, as a case study, the novelty of this research mainly includes two aspects. First, the vegetation type of this research is alpine swamp meadow which was little reported in previous studies about the vegetation degradation effects on soil hydraulic properties, especially Ks and FC. Second, we did find different variations characteristics of Ks and FC under alpine swamp meadow degradation and compared to previous studies, also quantitatively analyzed the influencing factors and mechanism. Detailed information could also be found in the revised manuscript. In addition, this study was based on the first-hand experimental data where the altitude of field sites is over 3,400 m with extreme cold environmental conditions. The data is not easy to get and relative precious and valuable for understanding the soil hydrological processes. So we believe that this study will be helpful to cognize the interaction between vegetation and soil moisture across different vegetation types and regions. Thankfully, we are pleased to see that the second reviewer gave this manuscript a positive judgment about the scientific significance. So, here we kindly ask you and the reviewers to reconsider the possibility of publishing this paper on HESS.

The point-by-point responses to the reviewers' comments are listed as following, and for specific revisions and changes, please see the revised manuscript appended.

**Anonymous Referee #1**
**Comment 1**The authors only gave several study cases in other Alpine mountainous regions, rather than deep review about the results of previous studies to extract the weakness and subsequently the motivations of this study.
**Response:** We basically agree the reviewer's comment. To be honest, the reviewing of previous studies was not adequate indeed. So, we revised and expanded the introduction section very carefully this time. Actually, we reviewed lots of related studies during this research, but didn't present them well in the paper. The references we reviewed include the studies on the influences of vegetation change on soil hydrological processes across different vegetation types and different regions around the world. After that, we deeply reviewed the related studies in alpine mountainous regions including the Rocky Mountain Range in North America, the Andean mountain range in South America, the Alps in Europe, Mt. Kilimanjaro in Africa, the Himalayans in Tibet, etc. We found that the researches on quantitative relationships between soil hydraulic properties and physiochemical properties are still relatively limited in alpine mountainous regions. We especially reviewed related researches in Tibetan Plateau which is the largest alpine plateau over the world. We found large discrepancies still exist in the obtained conclusions. Few literatures were reported to compare the different findings and explain the causes and mechanism. Furthermore, as a main grassland type in the eastern Tibetan Plateau, alpine swamp meadow is featured with unique

terrestrial-aquatic soil and vegetation characteristics. However, little attention has been paid to the effects of alpine swamp meadow degradation on soil hydraulic properties and the influencing mechanism. Therefore, based on the weakness we summarized from previous studies, the motivation of this research is to1) give a new case study of alpine swamp meadow about the vegetation degradation effects on Ks and FC, 2) compare different finding of Ks and FC responses to alpine vegetation degradation and explain the mechanism.

According to the analysis above, we re-write the $2^{nd}$, $3^{rd}$ and $4^{th}$paragraph of introduction section. Several previous studies were supplemented and deep reviewed. We also summarized the weakness of previous studies in each paragraph at the end.

In order to give a more clear impression of the review work we have done, we presented a table (**Table 1**) here to show the main references we reviewed and compare the main conclusions of previous studies. As the limited space for a formal paper, we didn't add this table to the manuscript. It could be easily found that up to now, conclusions about the effect of alpine meadow degradation on some key soil hydraulic properties, especially Ks, are not consistent. And the effects of alpine swamp meadow degradation on soil hydraulic properties were not reported.

Table 1 List and comparison of different previous studies reviewed in the manuscript

| No. | Study area | Vegetation type | Main conclusions | References |
|-----|-----------|-----------------|------------------|-----------|
| 1 | Himalayans | Grassland | Ks increased with the restoration of degraded grassland. | Ghimireet al., 2014 |
| 2 | Himalayans | Grassland | Ks increased with the restoration of degraded grassland. | Prasad et al., 2013 |
| 3 | Alps | Grassland | Grazing decreased FC and soil infiltration rate significantly | Leitinger et al., 2010 |
| 4 | Tibetan Plateau | Alpine meadow | Soil porosity decreased while Ks decreased firstly and increased later with degradation | Wang et al., 2014 |
| 5 | Tibetan Plateau | Alpine meadow | Soil moisture content, soil porosity and Ks decreased with degradation | Zeng et al., 2013 |
| 6 | Tibetan Plateau | Alpine steppe | Soil water holding capacity, soil moisture content decreased while Ks increased with degradation. | Wang et al., 2007 |
| 7 | Tibetan Plateau | Alpine meadow | FC increased firstly and then decreased, while Ks decreased firstly and then increased with degradation. | Wei et al., 2010 |
| 8 | Tibetan Plateau | Alpine meadow | Soil retention and soil moisture content decreased while Ks increased with degradation. | Wang et al., 2011 |
| 9 | Tibetan Plateau | Alpine meadow | Soil water holding capacity and Ks decreased with degradation. | You et al., 2015 |
| 10 | Tibetan Plateau | Alpine meadow | FC increased first and then decreased while Ks decreased. | Li et al., 2012 |

**Comment 2** The authors did not substantially highlight the new findings with respect to previous studies and address the reasons and mechanism for the differences. I found that most of the results are consistent or similar with the references.

**Response:** As mentioned above, the conclusions of grassland degradation impacts on soil hydraulic properties were not consistent. Our findings were based on the experimental data in alpine swamp meadow. Similar study reports in this vegetation type were not found. We also compared different conclusions across different vegetation types and regions. In fact, our new findings mainly include:1) the change patterns of FC and Ks with alpine swamp meadow degradation, and 2) the dominant factors of FC and Ks in degraded alpine swamp meadow. Our results showed that FC decreased consistently with degradation while Ks decreased firstly and increased later. Most previous studies reported Ks will continuously decreased with degradation. The amplitude of variation and change patterns were also different from the previous studies. Details could be found in revised manuscript. Besides, we applied redundancy analysis (RDA) method and found that capillary and non-capillary soil pores were the dominant factor of FC and

Ks, respectively. Moreover, we explained the reasons and mechanisms for the differences. For detailed corrections and revisions, please see section 4.2 of the manuscript.

**Comment 3** There was little detailed analysis about the effects of vegetation characteristics on soil properties.

**Response:** Thanks for pointing out the weakness of analyzing the effects of vegetation characteristics on soil properties. In order to address this issue, we selected VC (vegetation coverage), MB (above biomass), MA(above-ground biomass) as the vegetation indices. Then, we quantitatively analyzed the relationship between vegetation characteristics and different soil properties in 0-40 cm layers and 40-80cm layers respectively using Pearson correlation analysis (**Table2**). Corresponding contents were added in 3.3 (Results section) and 4.1 (Discussion section). Details could be found in the revised manuscript.

Table 2: Pearson correlation coefficient between vegetation characteristics and soil properties in 0-40cm and 40-80 cm layers

| Layers | Properties | BD | SOC | WSA | Sand | Silt | Clay | CP | NCP |
|--------|-----------|-----|------|------|------|------|------|-----|-----|
| 0-40 cm | VC | -0.710** | 0.769** | 0.747** | 0.533* | 0.472* | -0.491* | 0.829** | 0.155 |
| | MA | -0.811** | 0.899** | 0.902** | 0.838** | 0.698** | -0.735** | 0.808** | 0.345 |
| | MB | -0.635** | 0.860** | 0.800** | 0.672** | 0.646** | -0.662** | 0.615** | 0.028 |
| 40-80 cm | VC | -0.187 | 0.658** | 0.586** | 0.249 | 0.420 | -0.405 | 0.321 | -0.407 |
| | MB | -0.487* | 0.461* | 0.464* | 0.365 | 0.507* | -0.502* | 0.544* | -0.030 |
| | MA | -0.352 | 0.474* | 0.461* | 0.369 | 0.694** | -0.661** | 0.412 | 0.010 |

Note: *: significant at 0.05 level;**: significant at 0.01 level (2-tailed test); n=18.

**Comment 4** This point was well considered and addressed/
**Response:** Thanks for the approval.

**Anonymous Referee 2**

**Comment 1** Some paragraph of the English need to improve, and it needs help from native speaker.

**Response:** Thanks for pointing out the language problem. In order to improve the English, we invited Anna Herzberger (native American), also the coauthor of this manuscript, to polish the language throughout this manuscript again.

**Comment 2** "Monte-Carlo permutation test" was used in the manuscript, but the reference should be included

**Response:** Sorry for missing the reference. As the reviewer pointed out, we add the corresponding reference of "Monte-Carlo permutation test" in the revised manuscript.

**Comment 3** The format of Figure 4 should be checked.

**Response:** We have checked the format of Figure 4 and replaced with a reset one.

**Comment 4** The name of Axis of Figure 7 should be checked.

[revised manuscript text omitted]

---

## Author Response (AR3)

Dear editor,

Thank you for the supportive suggestions and comments on our manuscript. We were pleased to see that our manuscript has progressed to minor revision iteration since the last round of major revisions were mostly approved and recognized. As the anonymous reviewer #1 and you suggested, we added more discussion materials to highlight the shortcomings of this study. Besides, to improve the presentation quality, we invite professional language editing team to polish our language and the editing certificate was also appended.

The point-by-point responses to the reviewers' comments are listed as following. For specific revisions and corrections, please see the revised manuscript appended.

**Comment of Reviewer#1**

*The authors gave a deep review of previous studies in Alpine mountainous regions, and extracted two weakness and motivations of this study. First, there were large discrepancies about the effects of grassland degradation on soil properties, and the quantitative relationships between soil hydraulic properties and physiochemical properties were still relatively limited. Second, little attention was paid to the effects of alpine swamp meadow degradation on soil hydraulic properties. The authors also substantially compared the results with those in previous studies, and explained the reason.*

*My main concern is that the authors can discuss some further study scopes of the current study, for example, extending the study scale with more degradation gradients (there are only three gradients in this study), and discussing the interactions between grassland degradation and soil properties (not only the effects of grassland degradation on variations of soil properties ).*

**Response:**

*We thank for the approval of the major revisions and the suggestions with the discussion part. As the reviewer pointed out, the degradation classification of this study is somewhat rough, and the interactions between soil and vegetation should be further considered so as to fully understand the effect of alpine grassland degradation on soil hydrology. They are shortcomings of this research. Therefore, apart from the implications of this study, these uncertainties were added and addressed in section 4.3. The main content is as following.*

[revised manuscript text omitted]

**About the editor:**

| Editor | **Catherine Dandie** |
| --- | --- |
| | 2003 - PhD Environmental Microbiology - Flinders University |
| | *Microbiologist with interests in microbiological processes in soils* |
| | Full profile |

| Certificate issued by | Benjamin Shaw
Director

Liwen Bianji (Edanz Group China) |
[Figure]
 |
| --- | --- | --- |

While this certificate confirms the authors have used Edanz's editing services, we cannot guarantee that additional changes have not been made after our edits. It is the author's responsibility to ensure any unclear sentences in the manuscript are clarified for the Edanz editor.

Liwen Bianji (Edanz Group China)
Interchina Commercial Building, 1112A
No 33. Dengshikou Street, Dongcheng District, Beijing, P.C. 100006, China
Phone +86-10-6528-0877 Fax +86-10-6528-0834 Email editing@liwenbianji.cn